# Procalcitonin Evaluation of Antibiotic Use in COVID-19 Hospitalised Patients (PEACH): Protocol for a Retrospective Observational Study

**DOI:** 10.3390/mps5060095

**Published:** 2022-11-28

**Authors:** Joanne Euden, Philip Pallmann, Detelina Grozeva, Mahableshwar Albur, Stuart E. Bond, Lucy Brookes-Howell, Paul Dark, Thomas Hellyer, Susan Hopkins, Philip Howard, Martin J. Llewelyn, Wakunyambo Maboshe, Iain J. McCullagh, Margaret Ogden, Helena Parsons, David Partridge, Neil Powell, Dominick Shaw, Bethany Shinkins, Tamas Szakmany, Stacy Todd, Emma Thomas-Jones, Robert M. West, Enitan D. Carrol, Jonathan A. T. Sandoe

**Affiliations:** 1Centre for Trials Research, College of Biomedical and Life Sciences, Cardiff University, Cardiff CF14 4EP, UK; 2Severn Infectious Sciences, Southmead Hospital, North Bristol NHS Trust, Bristol BS10 5NB, UK; 3Medicines Optimisation and Pharmacy Services, Pindersfield Hospital, Mid Yorkshire Hospitals NHS Trust, Wakefield WF1 4DG, UK; 4Division of Immunology, Immunity to Infection and Respiratory Medicine, School of Biological Sciences, Faculty of Biology, Medicine and Health, University of Manchester, Manchester M13 9PL, UK; 5Critical Care Department, Royal Victoria Infirmary, The Newcastle-upon-Tyne Hospitals NHS Foundation Trust, Newcastle upon Tyne NE1 4LP, UK; 6Translational and Clinical Research Institute, Faculty of Medical Sciences, Newcastle University, Newcastle upon Tyne NE2 4HH, UK; 7UK Health Security Agency, London SW1P 3JR, UK; 8School of Healthcare, Faculty of Medicine and Health, University of Leeds, Leeds LS2 9JT, UK; 9Department of Pharmacy and Medicines Optimisation, NHS England, Leeds LS2 7UE, UK; 10Global Health and Infection, Brighton and Sussex Medical School, University of Sussex, Brighton BN1 9PS, UK; 11Perioperative and Critical Care Department, Institute of Transplantation, Freeman Hospital, Newcastle upon Tyne Hospital NHS Foundation Trust, Newcastle upon Tyne NE7 7DN, UK; 12Public and Patient Involvement Representative, NIHR, London SW1A 2NS, UK; 13Department of Microbiology, Laboratory Medicine, Northern General Hospital, Sheffield Teaching Hospital NHS Foundation Trust, Sheffield S5 7AU, UK; 14Pharmacy Department, Royal Cornwall Hospital, Royal Cornwall Hospitals NHS Foundation Trust, Truro TR1 3LJ, UK; 15Nottingham NIHR Respiratory BRC, University of Nottingham, Nottingham NG5 1PB, UK; 16Leeds Institute of Health Sciences, University of Leeds, Leeds LS2 9TJ, UK; 17Critical Care Directorate, Grange University Hospital, Aneurin Bevan University Health Board, Cwmbran NP44 2XJ, UK; 18Tropical and Infectious Disease Unit, The Royal Liverpool and Broadgreen University Hospitals NHS Trust, Liverpool L7 8XP, UK; 19Institute of Infection, Veterinary and Ecological Sciences, University of Liverpool, Liverpool L69 3BX, UK; 20Healthcare Associated Infection Group, Leeds Institute of Medical Research, School of Medicine, University of Leeds, Leeds LS2 9JT, UK; 21Department of Microbiology, Old Medical School, Leeds Teaching Hospital NHS Trust Leeds, Leeds LS1 3EX, UK

**Keywords:** antibiotics, COVID-19, antimicrobial stewardship, procalcitonin

## Abstract

Severe acute respiratory syndrome coronavirus 2 (SARS-CoV-2) is a novel virus responsible for the coronavirus disease 2019 (COVID-19) pandemic. Although COVID-19 is a viral illness, many patients admitted to hospital are prescribed antibiotics, based on concerns that COVID-19 patients may experience secondary bacterial infections, and the assumption that they may respond well to antibiotic therapy. This has led to an increase in antibiotic use for some hospitalised patients at a time when accumulating antibiotic resistance is a major global threat to health. Procalcitonin (PCT) is an inflammatory marker measured in blood samples and widely recommended to help diagnose bacterial infections and guide antibiotic treatment. The PEACH study will compare patient outcomes from English and Welsh hospitals that used PCT testing during the first wave of the COVID-19 pandemic with those from hospitals not using PCT. It will help to determine whether, and how, PCT testing should be used in the NHS in future waves of COVID-19 to protect patients from antibiotic overuse. PEACH is a retrospective observational cohort study using patient-level clinical data from acute hospital Trusts and Health Boards in England and Wales. The primary objective is to measure the difference in antibiotic use between COVID-19 patients who did or did not have PCT testing at the time of diagnosis. Secondary objectives include measuring differences in length of stay, mortality, intensive care unit admission, and resistant bacterial infections between these groups.

## 1. Introduction

Severe acute respiratory syndrome coronavirus 2 (SARS-CoV-2) is a novel virus currently causing a pandemic of the illness called coronavirus disease 2019 (COVID-19). Although the majority of patients affected by COVID-19 experience mild illness, a large number of people have been admitted to hospital and this continues to be the case [1]. Many patients have required oxygen therapy via positive pressure ventilation and some have required mechanical ventilation in intensive care [1]. SARS-CoV-2 is a virus and antibacterial agents (antibiotics) therefore have no direct killing effect on it. Despite this, many patients (45–100%) have been prescribed antibiotics [2,3,4,5,6,7,8]. Empirical antibiotic therapy is recommended in World Health Organisation guidelines for patients with suspected or confirmed severe COVID-19, COVID-19-related sepsis, and community and hospital-acquired pneumonia [1]. The evidence to support this practice is limited and current recommendations are based on concerns that patients may experience secondary bacterial infections that may respond to antibiotic therapy.

The COVID-19 pandemic, therefore, has the potential to drive an unnecessary increase in antibiotic use at a time when accumulating antibiotic resistance is also a global threat to health [9]. Antibiotic prescribing in patients who do not need antibiotics may drive excess mortality, for example through selection for resistant pathogens, *Clostridioides (Clostridium) difficile* infection and adverse drug reactions. There is evidence of an increase in the rate of antibiotic prescribing for patients hospitalised early in the COVID-19 pandemic [10]. Published data indicate that rates of secondary bacterial infection are low at 7–15% [2,6,7,11] and many confirmed secondary bacterial infections occur late in the illness; antibiotic use early in the course of COVID-19 may drive resistance in these later infections. Crucially, there is a big difference between the number of patients with secondary bacterial infection and those receiving antibiotics, particularly early in the course of infection, indicating that more studies on potential biomarkers are needed to guide appropriate antibiotic use.

Procalcitonin (PCT) is an inflammatory marker that can be measured in blood samples and is widely recommended to help diagnose severe bacterial infections and guide antibiotic treatment [12]. However, reviews of evidence to support its use in respiratory infections before the COVID-19 pandemic have found conflicting results [13,14]. Local guidelines were developed in many NHS hospitals, advising the use of PCT testing to assist in the decision to start or stop antibiotics in patients with COVID-19, but other NHS hospitals have not adopted this approach [15]. Local recommendations to use PCT have been pragmatic, in the absence of high-quality evidence in this clinical context, therefore the impact of its use requires evaluation. A recent analysis found that PCT had poor diagnostic accuracy for detecting microbiologically confirmed bacterial infection at the time of presentation with COVID-19 [16].

Large publicly funded randomised controlled trials (ADAPT-Sepsis, PRONTO [17] and BATCH [18]) are currently underway in the UK to assess the impact of PCT testing on antibiotic use but these are not specifically focused on COVID-19 patients. The first workstream of the PEACH project established that PCT was widely adopted in England and Wales during the first wave of the pandemic; use increased in intensive care units from 70 to 124/147 and in emergency departments and acute medical units from 17 to 74/147 [15]. It also established that introduction of PCT in English hospitals was initially associated with a fall in antibiotic use of around 1 defined daily dose (DDD)/patient/week, but as the first wave progressed, this effect was slowly eroded [19].

A number of centres have described their experience of using PCT during the pandemic, generally reporting that lower PCT values were associated with less antibiotic use [20,21,22,23]. Increased procalcitonin values have also been associated with clinical deterioration [24], so confounding by severity of infection is a concern in many of these studies, as well as their small sample size. The question remains as to whether PCT testing impacts antibiotic use, length of stay, intensive care unit admission, resistant infections and mortality in COVID-19 patients. Pending the findings from clinical trials, a multi-centre assessment of the utility of PCT testing in COVID-19, attempting to account for confounding factors, is needed to inform care in patients admitted to hospital with COVID-19 pneumonia, and to make interim recommendations using the best available evidence. Only observational (retrospective) and qualitative studies are open to us during this time-critical, recovery and learning period. We have therefore devised a mixed-methods approach to answer our research questions. Additional components of the PEACH study include qualitative analyses of interviews undertaken with health care workers (consultants, physicians, pharmacists and nurses) and health economics evaluation of the cost-effectiveness of the various PCT testing strategies employed by NHS Trusts and health boards during COVID-19. This protocol paper describes the patient-level data collection and analysis component of the PEACH study.

## 2. Aims and Objectives

### 2.1. Primary Objective

To measure the difference in early antibiotic use between COVID-19 patients who did/did not have PCT testing at the time of diagnosis.

### 2.2. Secondary Objective

To measure the differences in antibiotic use (total and ‘late’), length of stay, mortality (30 and 60-day), intensive care unit admission, and resistant bacterial infections between COVID-19 patients who did/did not have PCT testing at baseline; to assess the relationship between the number of PCT tests and total days of antibiotic therapy; to assess the relationship between C-reactive protein (CRP) and PCT to see if a low CRP predicts a low PCT; to describe the relationship between PCT and other commonly used inflammatory and infection markers (neutrophil count, white cell count); and to assess the relationship between tocilizumab and dexamethasone on inflammatory markers including PCT.

## 3. Methods

### 3.1. Study Design

This is a multicentre, retrospective observational cohort study using patient-level clinical data. This study has been designed with consideration of STROBE criteria [25].

### 3.2. Setting

Eleven NHS acute hospital Trusts and Health Boards from England and Wales, some of which did/did not use PCT routinely in COVID-19 patients during the first wave of the pandemic (Aneurin Bevan University Health Board, Leeds Teaching Hospitals NHS Trust, Liverpool University Hospitals NHS Foundation Trust, Mid Yorkshire Hospitals NHS Trust, Newcastle-upon-Tyne Hospitals NHS Foundation Trust, North Bristol NHS Trust, Nottingham University Hospitals NHS Trust, Royal Cornwall Hospitals NHS Trust, Salford Royal NHS Foundation Trust, Sheffield Teaching Hospital NHS Foundation Trust, University Hospital Sussex NHS Foundation Trust).

### 3.3. Participants

Participants who are eligible to be included in the data analysis will be identified from institutional databases and medical records by the clinical teams at each participating Trust or Health Board.

### 3.4. Inclusion Criteria

Data from all patients ≥16 years admitted to a participating Trust or Health Board and having a confirmed positive PCR COVID-19 test during the admission, between 1 February 2020 and 30 June 2020 are eligible to be analysed for this study.

### 3.5. Exclusion Criteria

Second and subsequent admissions after index admission with COVID-19 are to be excluded.

### 3.6. Outcome Measures

#### 3.6.1. Primary Outcome Measure

The primary outcome of this study is days of early antibiotics therapy (within the first 7 days after a first positive COVID-19 test sample).

#### 3.6.2. Secondary Outcome Measures

Total duration/length of antibiotic treatment;Duration (days) of late (>7 days) antibiotic treatment);Total DDDs of antibioticsDDDs of early (<7 days) antibiotic treatment;DDDs of late (>7 days) antibiotic treatment;Appropriateness of antibiotics according to local guidelines (% compliance), if practicable;30-day mortality;60-day mortality;ICU admission;ICU length of stay;Length of hospital stay;Antimicrobial-resistant (hospital-onset) secondary bacterial infection;Descriptive outcomes (e.g., types of antibiotic, route of administration and duration, frequency of PCT testing, types of secondary bacterial infection, relationship between different inflammatory markers and outcomes).

### 3.7. Data Variables

Routinely collected data from a patient’s episode of COVID-19 will be collected, in a standardised format into a dedicated study database. Some study variables may be more accurately recorded in primary care medical records (e.g., quality outcome framework registered conditions (comorbidity), body mass index, penicillin allergy records, ethnicity). We will attempt to obtain these specific variables through the linkage between secondary and primary care records coordinated at local collaborating centres. Where this is not possible, we will use comorbidities, body mass index, penicillin allergy records, and ethnicity recorded in secondary care records. Variables collected are listed in Appendix A.

### 3.8. Data Sources/Measurement

All data collection will be by electronic data capture using a bespoke database developed by the Centre for Trials Research (CTR) (Macro version 4.9.1, Ennov, Oxford, UK) and hosted by Cardiff University secure servers. It is encrypted and accessed by individual username and password. Data administrators or appropriately trained delegates will undertake the data collection. The variables constitute objective, routinely collected data from a patient’s episode of COVID-19 and each variable in the database will be collected in a standardised format. This will mean that the data for each variable from different centres can be validly compared in the analysis. Study data will be collected from patients’ notes with no patient interaction or consent obtained (see data management section below).

Day 1 of COVID-19 will be considered the day of the first positive sample; ‘early’ antibiotic use will be considered prescriptions on days 1–7, and ‘late’ after day 7. ‘Baseline’ variables will be those collected on/around the time of COVID-19 diagnosis, i.e., day 1 +/− 1

## 4. Data Management

The source data for PEACH will be collected from participants’ medical notes and NHS databases. Data will be de-identified at the source and assigned a unique identifier. The “master list” with identifiable data will be kept separately at individual centres, and used only to identify information from NHS databases (e.g., radiology, clinical chemistry and microbiology results). Only de-identified data will be collected and uploaded onto a central secure database and analysed.

Training for completion of the electronic case report form (CRF) will be provided to the appropriate study staff before study commencement. If missing or questionable data are identified, a data query will be raised on a data clarification form; this will be sent to the relevant site asking for clarification. The CRF pages will not be altered. All answered data queries and corrections will be signed off and dated by a delegated member of staff at the relevant site. The completed data clarification form will be returned to the CTR and an electronic copy retained at the site.

Details of data management procedures such as checking for missing, illegible or unusual values (range checks) will be specified in the PEACH Data Management Plan. Details of monitoring procedures will be specified in the PEACH Monitoring Plan.

## 5. Statistical Analysis

### 5.1. Sample Size

Data from around 7000 COVID-19 patients from 11 NHS acute hospital Trusts and Health Boards will be sourced, around half of which will have had PCT testing. Based on a minimally important clinical difference in antibiotic duration of 1 day (as proposed in the ADAPT-Sepsis trial https://warwick.ac.uk/fac/sci/med/research/ctu/trials/adaptsepsis/3.2_adapt-sepsis_trial_protocol__v3.0_12-feb-2019_clean.pdf (accessed on 24 August 2022)) between baseline PCT and non-PCT-tested patients, and a conservative assumption for the standard deviation (SD) of 6 days, 1500 matched patients will provide 90% power when using a two-sample *t*-test with two-sided 5% type I error rate.

### 5.2. Statistical Analysis

Descriptive statistics will be used for rates of PCT testing, antibiotic prescribing and secondary bacterial infection. This will be done overall and separately for those Trusts and Health Boards using/not using PCT, and also separately for patients who did/did not receive a baseline PCT test and a PCT test at any time. Comparative effect sizes, such as mean differences between groups, will be presented alongside 95% confidence intervals (CIs) wherever possible.

Multivariable regression models with random hospital effects will be used to examine factors affecting antibiotic prescribing including age, comorbidity, lung consolidation, secondary bacterial infection, CRP and PCT levels, and severity of illness (we will explore the use of CURB-65, qSOFA, and NEWS2 scores). Results will be presented as effect estimates with 95% CIs and *p*-values.

To assess the effect of PCT testing on patient outcomes and antibiotic use whilst ensuring an even distribution of important known confounders between groups, propensity score matching will be used. We will estimate a patient’s propensity for PCT testing with logistic regression on patient characteristics including age, sex, clinical severity of illness assessments, early secondary bacterial infection lung imaging findings, comorbidity and ethnicity. Baseline PCT and PCT at any time will be assessed; it is hypothesised that a baseline PCT may influence the duration of early antibiotic therapy while a PCT at any time may influence the duration of total or late antibiotic therapy. Patients who did/did not receive PCT testing will be matched without replacement with a 1:1 or 1:2 ratio according to their propensity. This will enable the comparison of several outcomes in between-patient groups which are balanced on important known confounders. Potential hospital effects will be accounted for in the model (e.g., by using random effects) or will be absorbed into the propensity scores. We will use a nearest-neighbour matching algorithm with/without a caliper. Alternative matching methods such as Mahalanobis distance matching and coarsened exact matching [26] or propensity score weighting [27] will also be explored. To check whether sufficient covariate balance has been achieved between the groups in the matched sample, Kolmogorov–Smirnov statistics and standardised mean differences will be assessed against thresholds of 0.1 [27]. The risk of unmeasured confounding will be quantified using E-values [28]

The primary analysis model for the propensity score-matched data will depend on the type of outcome: Poisson regression for counts (e.g., days on antibiotics), logistic regression for binary outcomes (e.g., ICU admission) and linear regression for continuous outcomes (e.g., DDDs of antibiotics). These will be adjusted for “truncation by death”, i.e., the problem that some patients die before another outcome (e.g., days on antibiotics) can be fully measured, thus leaving these outcome measures censored/undefined and with a seemingly better outcome (e.g., fewer days on antibiotics) due to the early death. To take this into account we will, in addition to a crude analysis restricted to the survivors in each group, perform a survivor average causal effect (SACE) analysis of the “always-survivors”, i.e., those who would have survived in either group [29].

Survival analysis will also be undertaken for outcomes that can be expressed as time-to-event (e.g., time until antibiotics are stopped) adjusting for confounders using a Cox regression if the proportional odds assumption holds, and after stratification otherwise. This will give greater power than the above analyses but requires further modelling assumptions. Importantly, it will allow us to perform competing risks modelling with death being a “competing risk”.

For all analyses, sensitivity analyses, including multiple imputations, will be undertaken to explore the impact of missing data. A detailed statistical analysis plan has been finalized and can be viewed in the Appendix A.

### 5.3. Minimisation of Bias

To reduce the risk of bias, consecutive patients fulfilling the eligibility criteria will be included. Identification of subjects will be carried out without the knowledge of outcomes and by separate teams from those carrying out the analysis. Potential confounding factors, i.e., those potentially influencing both the outcomes and the decision to use PCT testing will be discussed and agreed on in advance. Objective criteria for all study variables will be agreed upon in advance.

## 6. Patient and Public Involvement Statement

The research proposal and study protocol development have benefited from multiple interactions with Patient and Public Involvment (PPI) groups. Our PPI advisory panel, led by the PPI co-investigator, will lead on engagement with patient groups and the wider public through their involvement as members of the ICUsteps, Antibiotic Action (a public awareness group of the British Society for Antimicrobial Chemotherapy), and Antibiotic Research UK, and publicise the study through these channels. We will hold regular PPI focus group meetings and involve our PPI panel in all aspects of study design and dissemination.

## 7. Study Management

The study is sponsored by the University of Leeds and coordinated by Cardiff University CTR.

### 7.1. Study Management Group

The Study Management Group (SMG) will meet monthly throughout the study and will include the Co-chief Investigators, co-investigators, collaborators, study manager, data manager, study statisticians, qualitative researchers, health economists, and study administrator. SMG members will be required to sign up to the remit and conditions as set out in the SMG Charter.

### 7.2. Study Steering Committee

An independent Study Steering Committee (SSC) consisting of an independent chairperson, two independent members and a patient representative will provide oversight of the PEACH study. Members will be required to sign up to the remit and conditions as set out in the SSC Charter and will meet biannually.

## Data Availability

Not applicable at the time of publication. Data will be available upon reasonable request following the publication of the study results.

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
