# Peer review of "Procalcitonin Evaluation of Antibiotic Use in COVID-19 Hospitalised Patients (PEACH): Protocol for a Retrospective Observational Study"

_mps, 2022, doi:10.3390/mps5060095_

Round 1
Reviewer 1 Report
Please reconsider whether the study is retrospective as retrospective means presenting data from the outcome to exposure and dividing groups to ones with the outcome and without the outcome, i.e. a case-control study.
Some spaces are missing in text.
I am not sure of the relevance of publishing a study protocol for this type of study.
Author Response
Point 1: Unfortunately, we do not understand this comment. Our study design meets the commonly accepted definition of a retrospective cohort study (e.g., https://www.cancer.gov/publications/dictionaries/cancer-terms/def/retrospective-cohort-study). A detailed statistical analysis plan has been provided so we will leave it up to editors whether they wish to clarify with the reviewer.
Point 2: Spaces have been inserted into the manuscript where necessary.
Point 3: Retrospective, observational studies are at high risk of bias. By publishing a pre-specified protocol and statistical analysis plan, this risk is reduced and the findings made more valid. We have not made any changes in response.
Reviewer 2 Report
Dr Euden and colleagues are presenting a protocol for a retrospective cohort study on the use of procalcitonin and antibiotic use in patient with COVID-19 infection.
The topic is relevant as the virus is likely to remain a health care problem and antibiotic stewardship is very relevant.
The relevant sections of the protocol are well written and referenced. The statistical analysis plan is appropriate. The projected significant number of individuals will lend strength to the study while eliminating potential confounders.
Author Response
Many thanks for positive comments, we have not made any changes